# Integer Arithmetic Algorithm for Fundamental Frequency Identification of Oceanic Currents

**DOI:** 10.3390/s23146549

**Published:** 2023-07-20

**Authors:** Juan Montiel-Caminos, Nieves G. Hernandez-Gonzalez, Javier Sosa, Juan A. Montiel-Nelson

**Affiliations:** Institute for Applied Microelectronics (IUMA), University of Las Palmas de Gran Canaria, 35015 Las Palmas de Gran Canaria, Spain; juan.montiel@ulpgc.es (J.M.-C.); nieves@iuma.ulpgc.es (N.G.H.-G.); montiel@iuma.ulpgc.es (J.A.M.-N.)

**Keywords:** frequency parameters extraction, ocean tides and waves, underwater sensors, edge computing, offshore aquaculture infrastructures

## Abstract

Underwater sensor networks play a crucial role in collecting valuable data to monitor offshore aquaculture infrastructures. The number of deployed devices not only impacts the bandwidth for a highly constrained communication environment, but also the cost of the sensor network. On the other hand, industrial and literature current meters work as raw data loggers, and most of the calculations to determine the fundamental frequencies are performed offline on a desktop computer or in the cloud. Belonging to the edge computing research area, this paper presents an algorithm to extract the fundamental frequencies of water currents in an underwater sensor network deployed in offshore aquaculture infrastructures. The target sensor node is based on a commercial ultra-low-power microcontroller. The proposed fundamental frequency identification algorithm only requires the use of an integer arithmetic unit. Our approach exploits the mathematical properties of the finite impulse response (FIR) filtering in the integer domain. The design and implementation of the presented algorithm are discussed in detail in terms of FIR tuning/coefficient selection, memory usage and variable domain for its mathematical formulation aimed at reducing the computational effort required. The approach is validated using a shallow water current model and real-world raw data from an offshore aquaculture infrastructure. The extracted frequencies have a maximum error below a 4%.

## 1. Introduction

Nowadays, offshore energy and aquaculture infrastructures are hot topics. Their deployment and maintenance require knowledge of the marine conditions where they are installed. The monitoring of wind, waves, tides, and currents plays a crucial role [1]. In oceanographic literature, they are modeled using the mechanical wave theory. The monitoring of the traveling wave involves capturing and analyzing the temporal patterns of mechanical waves, including their frequency, amplitude, and other relevant characteristics [2,3].

The estimation and extraction of the traveling wave properties are a fundamental challenge in the field of signal processing, and in the analysis of time series data. This fundamental problem is defined as to accurately determine the frequency, amplitude, phase, and sometimes the damping factor of a composition of sinusoidal signals.

Furthermore, the extensive use of the edge computing paradigm defines a highly restrictive design scenario. This technique is based on performing most of the calculations where the data are acquired/generated, rather than in a centralized computing center or its distributed version called cloud computation. The application presented in [4] proposes a hierarchical computing structure using wireless sensor networks. This approach offers several advantages, such as fast response, effective data capture and retrieval, and accurate exception detection.

Despite the fact that all the approaches in the literature achieve their objective of extracting the frequency parameters, in most cases their implementation requires complex mathematical formulations and in all the cases the use of real or imaginary variables is mandatory. For example, the square root, sinusoidal, or division functions are implemented using mathematical software libraries and, by their nature, require high computational effort to solve. To implement a real number variable, the arithmetic unit must include a float processing unit (FPU) with, for example, double-precision or float registers [5]. Nowadays, the major challenge in embedded system applications is to reduce energy and power consumption; however, most of the published approaches involve specific hardware [6].

In this work, a frequency identification algorithm is proposed to be implemented in an integer arithmetic unit of a microcontroller for edge computation onboard the instrument presented in [7]. The purpose of this research is the design and implementation of an algorithm for a network of sensors attached to the mooring lines of an offshore aquaculture facility. This sensor network is used to monitor the forces present in such an infrastructure. Due to the large number of sensors and signals involved, this sensor network generates high raw data traffic. To address this challenge, an efficient solution is proposed through the practical application of the edge computing concept. Instead of sending all the raw signals generated by the sensors to the remote data processing center, the signal processing is performed on the instrument itself. This implies that data processing is performed on each sensor and only the relevant information and results are sent.

The use of this technique reduces the traffic load and optimizes the communication infrastructure. In addition, it allows faster decision making by processing data in real time, close to the generation source. Finally, it should be noted that this strategy increases the resilience of the system. In the event of network connectivity interruption, whether due to technical problems or adverse weather conditions, the sensors continue collecting and processing data autonomously.

In summary, the implementation of the edge computing concept in a sensor network in offshore aquaculture presents multiple advantages, such as a reduction in data traffic, faster decision making and greater system resilience. These advantages contribute to optimizing aquaculture production, improving operational efficiency and ensuring the well-being of farmed organisms in marine environments. The main contributions of this paper are as follows.
An integer arithmetic algorithm for the identification of fundamental frequencies oriented towards edge computing is proposed.The implementation of complex mathematical functions such as FIR filtering or derivation using only integer variables, additions and shift operations is studied in detail.The random nature of the traveling wave is evaluated based on the length of the acquired raw data.

This paper is organized as follows: Section 2 presents previous research about parameter extraction. In the next section, some necessary knowledge about ocean currents, instruments and the onboard sensors capabilities are presented. Section 4 introduces the proposed algorithm and explains the practical details to minimize the memory requirements, using only variables under the integer domain with additions and shifting operators. In Section 5, the proposal is evaluated and its results are discussed using shallow- and deep-water scenarios. Finally, the conclusions are presented in the last section.

## 2. Related Works

There are multiple approaches to obtain the frequencies involved in a signal. The most used is the fast Fourier transform (FFT) algorithm. Given a signal sampled in time, by applying this algorithm, it is possible to determine the involved frequency tones and their power spectral densities (PSD) [8]. Similarly, and based on the least squares algorithm, the authors in [9] estimated the harmonics and interharmonics for intelligent offshore microgrid (SMG) systems. The FFT usage and its window adaptation is the core of the approach.

Another solution to solve this problem is the proposed in [10] using an enhanced phase-locked loop (EPLL) system. For a distorted sinusoidal signal in a power system, they demonstrated that it is very robust with respect to noise and distortion due to disturbances and unbalanced system conditions. Moreover, in [11], an algorithm for estimating the harmonic components based on the demodulation approach using a controlled finite impulse response (FIR) filter is presented. The proposed technique may be a good candidate to replace the discrete Fourier transform (DFT) when the frequency of the electrical power system is sensitive to considerable variations.

A noteworthy technique in parameter estimation is the least squares optimization algorithm in combination with a Kalman filter [12]. This technique requires a large computational effort and is suitable for real-time applications [13]. For example, the authors in [14] monitored flicker and voltage fluctuations in the electrical power system by real-time implementation of the frequency-adaptive least squares Kalman technique. There are other heuristic approaches like the one presented in [15], where the authors proposed the use of a cuckoo search metaheuristic algorithm for the estimation of parameters in ocean surface wave modeling. In [16], an improved genetic algorithm for the estimation of parameters of sinusoidal signals was proposed.

Table 1 summarizes the main literature approaches to extracting the signal parameters. Most of the solutions require the use of Matlab. This is because this mathematical programming environment includes extensive function libraries for almost directly developing algorithms from the mathematical formulation. Of course, there are some proposals for implementations using non-desktop computers. However, the embedded systems used are high-end digital signal processors (DSPs) suitable for environments where there are no power consumption restrictions. For example, the TMS320C6711 used in [10] has a working frequency of 250 MHz and requires 0.87 mA to operate.

## 3. Background

Before continuing with this investigation, it is necessary to clarify some concepts related to the nature of the measurements and instruments.

### 3.1. Ocean Currents

The target of the final application is to monitor ocean currents in an offshore aquaculture infrastructure. This research uses the instrument introduced in [7], and this instrument is deployed following the schema described in [17]. Each installed instrument is located on the mooring line of an offshore aquaculture infrastructure and is designed to make long-term measurements of marine currents in deep waters. It should be noted that the instrument is based on the principle of tilt-drag. Through this principle, the tilt angle is measured, which is a function of the speed of the water in the location where the device is installed.

To achieve this, it is necessary to make considerations that have been validated in the oceanographic literature. It is known that the ocean surface is composed of a succession of maxima and minima with an irregular distribution in time, although it is described in terms of its temporal regularity [18]. Furthermore, regardless of depth or weather event, waves can be described using the Fourier model [19].
(1)$$\eta \left(t\right)=a_{0}+\sum_{i=1}^{N-1}a_{i}sin\left(\omega_{i}t+\phi_{i}\right),$$
where $a_{0}$ is the averaged water level at the ocean surface; $a_{i}$, $w_{i}$ and $\phi_{i}$ are the amplitude, frequency and phase for each sinusoidal component *N* of the model, respectively. The simplest critical corner of this model is the one that describes a regular and therefore periodic wave, which corresponds to a single sinusoidal function. Since ω=2πf, the period of the ocean waves is T=2π/ω.

The application of the Fourier model allows to analyze and understand the time dynamic characteristics of the waves, which is relevant for the measurement and calculation of marine currents. Considering these characteristics and using the proposed algorithm, we sought to obtain precise estimates of the marine currents based on the tilt angle measurements made by the instrument on the mooring line.

It is important to note that the gravitational forces exerted by the Sun and Moon are constantly present. Any body of water, such as oceans and seas, is affected by these attractive and repulsive gravitational forces. Gravity causes the movement of ocean water to restore gravitational balance. In the case of deep water, this movement follows a harmonic behavior. At regular intervals of 6 h, 12 min, and 30 s, a maximum or minimum value occurs at the ocean surface, known as high tide and low tide, respectively. In terms of the Fourier model of Equation (Equation 1), the period of the gravitational wave, denoted as *T*, has a value of 12.417 h.

At the ocean surface, one of the components of wave generation is the wind. The effects of surface forces are attenuated as a function of depth. Thus, to define deep water, the literature considers that the influence of waves generated by wind is negligible from a depth $d_{n}$ equal to or greater than half the wavelength $l_{w}$ of the surface waves, following the equation:(2)$$d_{n}=l_{w}/2$$

In other words, the behavior model of shallow water depends on the seabed and weather conditions, such as wind and temperature, among others, in addition to the gravitational forces of attraction. In the case of deep water, the behavior mainly depends on gravitational forces, temperature and salinity. In general, the research literature considers deep water between 7 and 10 m. Finally, the ocean is always moving following the theory of ocean circulation defined in [20].

### 3.2. Ocean Current Meters

The most common ocean current meters are based on the Doppler effect or the tilt principle. The cost of the instrument is mainly due to the implemented methodology. Despite its accuracy and from an economic point of view, the deployment of several Doppler instruments around an offshore aquaculture installation is prohibitive. The cost for a single Doppler instrument is above 8K USD. Due to their low cost and easy maintenance, tilt-based instruments are the most widely used. Table 2 compares several ocean current meters from the research literature and commercial products in terms of the used method, deployment location and cost.

In general, regardless of the methodology used, the monitoring approaches in the literature and industrial solutions follow the scheme presented in Figure 1. The acquired data obtained by the ocean current meter reaches the monitoring center via wireless communication. The wireless communication module of the instrument is placed at the ocean surface level on a buoy. The ocean current meter is wired directly to the communications module. An anchor on the seabed is necessary to avoid movement of the instrument due to displacements of the buoy.

From a measurement point of view, the communications bandwidth is not restricted because underwater transmissions are carried out through cables. If each deployed ocean current meter requires its own cables, the maintenance/operational costs of the aquaculture infrastructure increase. A solution to reduce the cost is to use a single cable instead multiple. However, bandwidth constraints arise as more and more sensors are deployed.

### 3.3. Onboard Instrument Capabilities

In addition to classical characteristics such as precision and accuracy, oceanographic instrumentation is subject to a severe set of requirements and specifications. The weight and size of the batteries define a dimension limit for the instrument. However, there is a trade-off between battery size and available energy. Today, the research literature and commercial ocean current meters follow the same philosophy, that is, to minimize energy requirements. In terms of ultra-low-power designs, the use of a hardware floating point unit processor increases the power consumption by at least twenty-five times. This is the case, for example, when using an ARM M4 architecture instead of an ARM M0+ architecture using the NXP Kinetis development platform [30,31]. In this sense, the design of the instrument is mainly focused on acquiring raw data, and the data processing is located outside the instrument to reduce power consumption.

Another characteristic is the reduced execution speed of operations, i.e., this type of microcontroller has a low operation frequency. The linear dependence between frequency and power consumption involves that to minimize the energy consumption, the execution speed must be reduced. Moreover, the available memory is limited in comparison with general purpose microprocessors. Instead of several gigabytes of RAM, ultra-low-power microcontrollers have several kilobytes of RAM. In this type of systems, the RAM is used to implement the software variables. The embedded application is stored and ran on flash memory, which is no larger than a couple of kilobytes. Note that in most of the cases, the microcontroller is based on a specific bare metal software. That is, there is no general operating system. In essence, bare metal serves as a basic operating system that is closely tailored to meet the execution demands of the final embedded application.

For example, tilt-based instruments at least provide measurements of acceleration using microelectromechanical systems (MEMSs) and orientation of the equipment with a compass. This information is called raw data. Doppler solutions detect the frequency variation of the emitted signal and provide measurements for each observation cone. In addition, the instrument provides orientation using a compass and also its inclination from an MEMS. Table 2 listed the main features of current ocean current meters.

## 4. Method

### 4.1. Parameters Extraction

In order to model a target application, a parameter extraction procedure is basically a technique to obtain characteristic values from observed phenomena [32]. Each specific application defines its own group of parameters. As base requirement in all the cases, it is mandatory to obtain the fundamental frequency and its amplitude. Some applications are interested in obtaining the sequence of frequencies.

In our case, due to the ocean current’s physical limitations, not all frequencies are possible. In this sense, it is possible to remove the noise and other non-interested signal components from the sampled data. For example, the current components due to the Sun and Moon’s gravitational attraction have a period close to 22 μs. However, on the ocean surface we can only consider frequencies in the range of no more than a couple of Hz. If the measurement target is to observe the Hz range, the gravitational tide components are reflected in the measurement as a continuous value/offset. On the other hand, it is well known that given a signal in the time domain, the resolution of its representation in frequency depends on two factors. Obviously, the first one is the sampling frequency and the other is the size/length of the sampled data. Both solutions have their own consequences. Of course, increasing the sampling frequency also increases the number of samples, thus improving the reconstruction of the signal in its temporal or frequency representation. However, it also increases the cut-off frequency of the low-pass filter of the instrument. Therefore, the expansion of this limit introduces noise in the sampled data. The higher the sample rate, the greater the noise introduced.

Furthermore, the frequency resolution of the signal increases if the length of the studied data increased. In this case, the equivalent low-pass filter keeps the same cut-off frequency. Increasing the length of the sampled data without changing the sampling frequency basically extends the time of the signal observation. This methodology is ideal when the observed signal is time-invariant. However, a simple observation of ocean currents reflects temporal variations in wave shapes. In same way, the camera exposure parameter in a photo, increasing the sampled time period (acquired data length) could result in overlapping signals in their frequency representation.

Based on the previous paragraphs, we conclude that the frequency representation of deep-water ocean currents could not be a solution to model these physical phenomena. Its evaluation in the time domain is more convenient to avoid, for example, temporal overlapping. In addition, currently used simulation environments use a time domain specification as input parameters.

### 4.2. Proposed Algorithm

Figure 2 presents our algorithm for obtaining the fundamental frequency of the acquired data. It is composed of four phases, which are a band-pass filter, a phase remover, a derivation and a detector.

The first stage, as a band-pass filter, removes the non-interested signals. Then, the second stage eliminates the phase information. The third stage derives the signal to obtain the sequence of maxima and minima of the acquired data. Finally, the last step detects the peaks, valleys and zeros of the signal. The first step has an unique and clear objective; however, the second step has two missions. A goal to convert each minimum in the signal to a maximum. Using this procedure, the distance between two maxima in the resulting signal is half the period (T/2) of the signal being analyzed. The second goal is to change each zero-crossing point of the data collected into a minimum in the resulting signal.

The target of the last two states is to locate the mentioned maximum and minimum values through the signal trend. In this way, the derivative function in the third stage calculates the trend of the signal. Finally, the last stage detects the peaks and valleys in the trend. The algorithm produces a vector of consecutive maxima and minima in the acquired data using the maxima of the trend. On the other hand, our proposed algorithm generates a vector of the zero-crossing points in the acquired data based on the location of the trend minima. The obtained maximum and minimum values are temporally and sequentially separated by a quarter period of the original signal. The proposed block diagram shown in Figure 2 is composed of well-known signal processing functions. However, its low-level implementation in the integer arithmetic and demonstrating its usefulness are challenging tasks.

### 4.3. Practical Implementation

#### 4.3.1. FIR Filter

The moving average (MA) filter is a signal processing method that contributes to reducing the impact of noise or variations in a time-series signal. In other words, it is basically a low-pass finite impulse response (FIR) filter. To accomplish this signal processing technique, a mean value is calculated over a defined time window for the sampled signal. Then, at each time interval, the original signal value is substituted with the calculated average value.
(3)$$Y\left[n\right]=\sum_{i=0}^{N-1}k\left[i\right]\cdot x\left[n-i\right],\;\;\;\;\;\;\;k\left[i\right]=\frac{1}{N}$$

As a formula, it is quite simple. It is composed of N−1 summations of *N* multiplications and *N* divisions. However, this simplicity becomes a nightmare when it has to be implemented into current ultra-low-power microcontrollers. One disadvantage of this type of processor is that it lacks a floating point unit. If it is necessary to implement it onboard as specified, the entire process must be carried out using software libraries. On the other hand, despite a multiplication unit existing in most of the cases, the division is performed by a software algorithm.

Taking in consideration that the acquired data are expressed as a sequence/vector of integer numbers, a basic idea is to keep this data and the applied algorithms in the same format. The filter coefficient k[i] is never an integer. However, by correctly selecting the coefficients, the division can be transformed into an arithmetical shift operation (e.g., ASR assembler operator in ARM M0+ [33]) that is available in every integer processing unit. In our case, we chose that the division factor (*N*) must be a power of 2.

Figure 3 shows the frequency response of the proposed FIR filter when the coefficients are 4, 8, 16 and 32 for a sampling frequency of 1 Hz. Since the vertical axis is expressed in terms of decibels (dB), the half power level (−3 dB) is also expressed in the graph.

The half powers for a normalized frequency of 1 Hz are 0.308, 0.151, 0.075, and 0.038 Hz for those coefficients. However, as observed in the figure, the frequency response corresponds with a low-pass filter. In our application, we are interested in a band-pass filter. The solution is to then apply two FIR filters using the power of two coefficients in the way that it is shown in Figure 4.

In Figure 4, the upper filter FIR${}_{A}$ has a lower coefficient than FIR${}_{B}$. In our application, we chose these coefficients to be twice each other. Our aim is to minimize the number of mathematical operations. In this sense, in terms of the computational effort measured in the required execution time, the selected coefficients are increasingly optimal as they become closer. Given the pair of coefficient ratios, a set of band-pass filters are obtained, i.e., their cut-off and band-pass frequencies. Table 3 presents the upper ($F_{h}$) and lower ($F_{l}$) cut-off frequencies of the obtained band-pass filter using the scheme presented in Figure 4 for a sampling frequency of 1 Hz.

The frequency response of the proposed band-pass filters set is shown in Figure 5a. The graph presents the filters for $N_{A}$ equal to 2 up to 19 with unit increments. The filter on the right corresponds to $N_{A}$ equal to 2, and the one on the left to $N_{A}$ equal to 19. In terms of attenuation, it is noteworthy that the filter decays very quickly after the cut-off frequencies. However, other gain peaks appear as the frequency increases. In general, the first undesirable peak is at 6 dB, while the others have greater attenuation.

Since the FIR filter is a linear function and the coefficients are selected as a potential function, we can easily determine the cut-off frequencies and obtain the bandwidth of the implemented band-pass filter as a closed form of an analytical function. Figure 5b shows the upper (Fh) and lower (Fl) frequencies, and the bandwidth (BW) as a function of the coefficient.

The formulation obtained is:(4)$$F_{h}\left(N\right)=\frac{1}{2^{0.99622N-0.38614}}\left(\mathrm{Hz}\right)$$
(5)$$F_{l}\left(N\right)=\frac{1}{2^{0.98070N+2.17562}}\left(\mathrm{Hz}\right)$$
(6)$$BW\left(N\right)=\frac{1}{2^{1.00027N-0.12364}}\left(\mathrm{Hz}\right)$$

The first approach to implement the FIR filter is quite direct and simple. Since the input data are an array of integers, the implementation is made using a loop to compute the summation. This direct solution has a main implementation problem. Because the microcontroller memory is very limited in size, as soon as the FIR coefficients increase, the data length of the array exceeds the memory size. Of course, in most cases, the raw data to be processed is also stored/recorded on external memories such as secure digital (SD) or microSD cards. It is possible to use these external devices and process the raw data in this way. However, the power consumption required to access external memories makes it limiting in practical ultra-low-power applications.

The proposed band-pass filter ($FIR_{BP}$) presented in Figure 4 is expressed as:(7)$$FIR_{BP}^{\prime }=FIR_{A}-FIR_{B}$$
where:(8)$$FIR_{A}\left[n\right]=\frac{1}{N_{A}}\sum_{i=0}^{N_{A}-1}x\left[n-i\right],\;\;\;\;N_{A}=2^{A}$$
(9)$$FIR_{B}\left[n\right]=\frac{1}{N_{B}}\sum_{i=0}^{N_{B}-1}x\left[n-i\right],\;\;\;\;N_{B}=2^{B}$$
and
(10)B=A+1

In this formulation, we assume that the lowest index in the acquired raw data array (x[n]) corresponds to the newest temporal sample compared to the highest value representing the oldest sample. In this scenario, where the FIR coefficients are twice the other (see Equation (Equation 10) and Figure 6a,b), the band-pass filter can be rewritten as:(11)$$FIR_{BP}^{\prime }\left[n\right]=\frac{1}{2N_{BP}}\left(\sum_{i=0}^{N_{BP-1}}x\left[n-i\right]-\sum_{i=N_{BP}}^{2N_{BP}-1}x\left[n-i\right]\right),\;\;\;\;N_{BP}=2^{BP}$$

The implementation of this formulation produces a great advantage over the previous one. First at all, the calculation of the FIR${}_{B}$ filter requires half the elements. In the same way, the complete procedure requires a third less memory.

#### 4.3.2. Algorithm Implementation

Figure 7 illustrates the implementation of our proposed algorithm in pseudocode. Obviously, the input is the acquired raw data and the filter coefficient. The output is an array with each peak, valley and zero of the processed raw data. Finally, the number of these events is also provided. In order to clarify this explanation, we labeled each representative line of our pseudocode.

The band-pass filter is implemented on lines L02 to L18. The solution adopted was performed in three phases. At the beginning, from L02 to L05, the first sum of the band-pass FIR is calculated for the initial $N_{A}$ samples of the incoming data. In the second phase, the other summations are obtained on lines L06 to L11. Note that this realization follows the procedure presented in Figure 6c, where each FIR of length $N_{A}$ is computed before continuing with the rest of the formula implementation. There are two important issues on these lines. The first is the variable “tdata”, containing the last summation of the input data of the FIR of size $N_{A}$. Each new summation is calculated using the previous one, adding the new incoming sample and removing the oldest sample from the old summation. In this way, the computation of all FIR of size $N_{A}$ requires only *L* additions and L−1 subtractions, where *L* is the length of the incoming raw data to be processed.

On the other hand, the weakness of this solution comes from its own temporary variable “tdata”. Its size, in terms of bit width, determines the maximum number of elements that can be added, and therefore the maximum value of $N_{A}$. In general, we assume that incoming data are generated using a 16-bit analog-to-digital converter (ADC). If the temporal variable “tdata” is defined as a 32-bit integer, it is possible to calculate $2^{16}=65,536$ additions.

Finally, the band-pass filter $FIR_{BP}$ is computed on lines L12–L18. The use of an arithmetic shifter as a divider is located in line L14. In addition, the absolute value of the filtered data is also implemented in this loop in lines L14–L17 (see Figure 2 for more details). Then, lines L19–L25 derive the filtering and unsigned resulting data. This process consists of subtracting the data between them. For each computed value, its temporary predecessor is subtracted. Since we are interested in detecting the maximum and minimum locations of the absolute value of the filtered incoming signal, we focus the search algorithm on the slope change. Furthermore, this is the reason why we only store this parameter. The result is an array containing a −1 when the incoming signal has a negative slope and a 1 otherwise.

In this scenario, the peak detector operates as our slope’s trend representation to determine any change in tendency by subtracting to each element its predecessor. In other words, we derive the slope’s trend array. If the derivation produces a coefficient of 2, this means that we are detecting a zero value in the original incoming raw data signal. If the obtain a value of −2, the detection is a maximum or a minimum.

As stated in previous paragraphs, the output of the proposed algorithm is an array with the locations of the maxima, minima and zeros indexed. Since the target of the algorithm is to extract the fundamental frequencies that are in the selected band-pass region, we do not provide their real values. We only determine their temporal locations.

## 5. Experiments and Discussion

Now we check the usefulness of our proposed algorithm. The simplest test is to use the sampled signal as a pure sinusoidal function. However, the nature of ocean currents do not generate a pure sinusoidal tone. In this sense, the algorithm is evaluated using a well-known shallow-water model and real raw data from a deep-water current meter installed on an offshore aquaculture infrastructure.

In addition, since the aim of our algorithm is to extract the frequencies involved in the incoming signal, the magnitude meaning of the measure does not care. From another point of view, the raw data acquired are basically a sequence of integers. Converting this sequence of integer values to a sequence of values converted to the target measure’s units does not provide any additional information to the instrument. Moreover, the conversion complicates the raw data computation by changing its domain to that of real numbers. This is the reason for not including units in all the amplitudes in presented figures and tables.

### 5.1. Shallow Waters

#### 5.1.1. Setup

A classical synthetic signal used as a test for the parameter extraction of continuous functions and to characterize shallow water behaviors is the called F3 function [15]. This signal is defined as:(12)$$f\left(t\right)=-\sum_{j=1}^{5}\left[j*sin[(j+1)t+j\right]]$$

It is composed of five sinusoidal functions. The F3 function is presented in Figure 8a in the time domain. In this corner case, we set a sampling frequency of 12.5 Hz. The length of the acquired raw signal is 256. Therefore, the total time processed is 20.48 s (see Figure 8a). Furthermore, the frequency representation of the signal and the used FIR filters are also presented in Figure 8b.

We observe the five sinusoidal tones located below 1 Hz. They are located at 0.342, 0.488, 0.635, 0.781 and 0.977 Hz. The band-pass FIR filter used is based on $N_{A}=3$. This is the summations of the lower coefficient FIR filter with only eight additions. Although the signal is below 1 Hz, the implemented band-pass filter does not focus on it. However, there is no other signal or noise above 1 Hz. Finally, the F3 signal does not include a continuous level/offset.

#### 5.1.2. Evaluation

To understand the frequency extraction process, we present the obtained data for each meaningful step of our proposed algorithm. Figure 9 shows the evolution of the data computed through the execution of the algorithm. Note that Figure 8 represents the F3 signal in the continuous time domain. This signal is acquired using a 16-bit analog-to-digital converter (ADC) in this corner case. In addition, we assume a full-scale ADC of [+15, −15] units.

Figure 9a presents the F3 signal filtered using a FIR filter with $N_{A}$ and $N_{B}=N_{A}$ + 1 coefficients. They are labeled as FIR${}_{A}$ and FIR${}_{B}$, respectively. Figure 9b depicts the results of the applied band-pass FIR filter (FIR${}_{BP}$). Then, the absolute value of the FIR${}_{BP}$ is presented in Figure 9c. In Figure 9d, a previous intermediate result is derived to obtain the peaks, valleys and zeros from the original F3 signal. Figure 9e shows the array which contains the slope trend of the previous derivation. Finally, the last graph illustrates the values of the obtained detection array. In Figure 9f, a −2 value means a maximum or minimum has been detected, and 2 represents a zero-crossing point in the F3 signal.

At this point, it is mandatory to remember that the reconstruction of the studied signal is not the objective of this research. However, to evaluate the extraction process and clarify the usefulness of the obtained values, we have performed some reconstruction for this purpose. In this sense, the output vector of the peaks, valleys and zeros location is depicted in Figure 10 as vertical lines overlapping the sampled and processed F3 signal.

The algorithm is capable of finding the target locations in the F3 signal. Of course, the initial and final samples do not comply with this. From an application point of view, there is an implementation dilemma when applying the proposed algorithm. The key question is the latency of the algorithm. In order to obtain the first value of the proposed band-pass filter, $N_{A}$ sample are required.

In practical applications, the acquisition system samples the measured variable for a while, and then the acquired raw data are stored, processed, or both. In case of continuous acquisition, the measured variable is stored, for example, in a FIFO and the real-time application specification determines the period between the raw data calculations. In both cases, the algorithm produces a latency at the beginning and end of the process.

To minimize the impact on algorithm latency, we insert $N_{A}$ samples with zero values at the beginning of the acquired raw data. Similarly, we insert the same number of zero value samples at the end of the raw data to be evaluated. Therefore, the initial calculations of FIR${}_{A}$ and FIR${}_{B}$ are not reliable, and as soon as the first samples of $N_{A}$ are processed, the result converges to the correct solution of our proposed algorithm.

Based on Equation (Equation 11), this assumption implies that the values of the samples from $N_{A}$ + 1 to $2N_{A}$ are zero in first FIR${}_{BP}$ computation. Therefore, the band-pass filter FIR${}_{BP}$ is initially equal to the average of the raw signal for the values of $N_{A}$. As soon as the processing continues, the filter output approaches its correct output value.

Therefore, the result of preceding and succeeding with $N_{A}$ zero values in the raw data are that the filtering follows the input signal at the beginning and end of the processing. We observe this effect in the left and right of Figure 9b. This behavior is not arbitrary. From a practical point of view, the insertion of samples with zero values is equivalent to having an empty FIFO that fills or empties with the acquired data as the acquisition begins or ends.

In addition to the latency due to the FIR${}_{BP}$, there is another minor problem related to the indexing of samples throughout the calculation process. In our algorithm, each derivative calculates a new value that corresponds to two contiguous samples in time. Therefore, the computed slope is temporally shifted from the original data by half a sampling period. Because the proposed algorithm applies two times the derivation function, the final array with the index of peaks, valleys and zeros has an offset of one unit.

Figure 10a presents the acquired F3 signal processed with our frequency extractor algorithm. Its right axis represents the 16-bits integer value of the samples and the horizontal axis presents the sample index of the signal. In addition, the locations obtained with our algorithm are marked using vertical lines. Continuous lines identify the detected zeroes and the discontinuous traces are the maxima and minima.

In order to avoid the initial and end behavior due to the FIR${}_{BP}$ function, that is, the latency, we start to represent the location vector after the first zero is found. The offset problem is also taken into consideration in this image. We observe that the algorithm correctly finds all the peaks, valleys and zeros.

On the other hand, Figure 10b presents the original acquired raw data for the F3 signal and the sinusoidal reconstruction of the signal based on the output index array obtained with our proposed extraction algorithm. In this case, our reconstruction assumes that between two zeros there is a half sinusoidal signal. The maximum of this half sinusoidal signal is defined by the value of the sampled signal between the zeros.

Although the reconstruction of the signal is not studied in this research, the comparison between the original raw data and the synthesized data based on our frequency extraction algorithm is highly accurate. In particular, the zero-crossing points of the F3 signal have a negligible error. The error in the peaks and valleys is less than 7% in the worst cases. This error is located in some of the peaks and valleys in the signal. The average error is less than 2.5% in the rest of the reconstructed F3 signal. The main reason for these errors, in addition to the selected reconstruction methodology, is the signal sampling. If sampling a continuous F3 signal does not sample close to the peaks and valleys, its reconstruction increases the error.

Figure 10c represents the sampled F3 signal in the frequency domain and the frequency representation for the reconstructed signal. The graph only presents signals below 1 Hz. Higher frequencies do not include any information, that is, the amplitudes for those frequencies are close to zero. In this figure, we observe the correct alignment between the sampled signal and the reconstruction. Furthermore, the in frequency response, the obtained results compared with the original data has an error of less than 2%.

### 5.2. Deep Waters

#### 5.2.1. Setup

In this corner case, we use real data acquired from an aquaculture infrastructure in the Canary Islands. The measurements were performed from May 2022 to September 2022 over 180 days using the deep-water current meter presented in [7]. The acquired raw data consists of the measured acceleration of the submerged device at a depth of 15–18 m. Although each water current velocity sample is decomposed into three orthogonal axes, this research only focuses its processing on the z-axis that is parallel to the gravitational acceleration vector.

The acquisition system is based on the MMA8451Q microelectromechanical system (MEMS) device to measure tri-axial accelerations. Its ADC includes its own first in first out (FIFO) memory for 32 samples and the measurement were performed with 14/8-bit resolution. In our case, the acquired raw accelerations have 14 bits. The whole system is programmed to measure at a maximum of 4096 samples. Because the sampling frequency is set to 12.5 Hz, the raw data represent a period of 327.68 s (5.46 min).

#### 5.2.2. Evaluation

Figure 11 presents the results of applying our proposed frequency extraction algorithm to the raw data captured from the described instrument. Each row in the figure depicts different capture periods. The first row is a 128-sample-length signal. The second one has 256 samples. Each new row duplicates the length of the previous one. Finally, last row shows a signal with 4096 samples. In terms of time, the first row represents a sampling period of 10.24 s, while the last one is 327.68 s long.

On the other hand, each row in Figure 11 contains the time and frequency domain representations of the processed raw signal. On each row, the first column shows the temporal representation of the processed raw signal and the second column its frequency domain equivalent. However, to clarify the time domain representation, only the last ten seconds of the raw signal are shown. In addition, the frequency representation is below 2 Hz despite its bandwidth being 6.25 Hz. The signals above 2 Hz are negligible.

In Figure 11, the time domain representation of the processed raw signal is overlapped with marks locating the peaks, valleys, and zeros found by our proposal. In the same way as previous F3 signal, here a detected zero is identified with a vertical continuous line and a peak or valley using a discontinuous line.

The left vertical axis of the time domain representations show a normalized value from zero to one. The right vertical axis shows the real values obtained from the ADC from the MEMS device. The conversion factor between the acceleration-measured variable and the integer value from the ADC is 9.81/4096 (m/s${}^{2}$). In addition, the frequency representation shows the original raw signal and reconstructed signal using the parameters extracted with our proposed algorithm assuming a model between zeros as a half period sinusoidal signal.

Since the sampling frequency is fixed by the instrument at 12.5 samples per second, the resolution in terms of frequency depends on the number of samples in the raw data to be processed. The resolutions for a representation using 128, 256, 512, 1024, 2048 and 4096 samples are 0.1, 0.05, 0.025, 0.0125, 0.00625 and 0.0031 Hz, respectively. As a consequence, the resolution of the results increases as the length of the samples to be processed is greater.

In Figure 11, at first glance, the raw signal is not symmetrical. Although it is not a repetitive signal, there are some similar and contiguous patterns that appear sporadically. This is normal behavior of the measured variable for a ocean deep-water current. Looking at the frequency representation, we can identify two frequency tones in Figure 11(a.2). Since the first row in this figure is the lowest resolution, the next two rows show both frequency tones in close proximity with greater accuracy (see Figure 11(b.2,c.2) for more details).

However, the two highest-resolution results shown in Figure 11(e.2,f.2) do not clearly show both frequency tones. Only the 0.147 Hz frequency tone can be estimated. The power of the tones found with lower resolutions has been homogenized with the powers of nearby frequencies. From a practical point of view, this behavior can be explained in two ways.

The first is based on the Fourier transform. It is well known that given a sampling period T, the energy of each tone is calculated in the time lapse −T/2 to T/2. This time lapse is equivalent to the resolution bandwidth of the spectrum analyzer (RBW). As the measurement filter, the captured energy is greater the wider its bandwidth. Our sampling frequency set by the instrument is 12.5 Hz. However, increasing the number of samples computed is equivalent to increasing the sampling frequency.

The second effect is due to the practical consequence of increasing the number of samples processed while keeping the sampling frequency unaltered. In this scenario, the observed time increases. However, the signal is not periodic as we stated previously, and the ocean deep-water current presents some patterns that appear randomly. Therefore, the frequency representation obtained with the largest number of samples is equivalent to a long exposure photo of the frequency domain.

As a conclusion, a greater number of samples introduces more and more errors in the reconstruction of the signal if performed based on the frequency response.

Table 4 shows the errors for the reconstructed signal using the extracted frequency parameters in comparison with the acquired raw data from the instrument. In addition, the number of zeros, peaks and valleys (P and V) are detailed. This table was obtained after applying our proposed algorithm to 189 days of data.

The error is smaller the smaller the size of the processed data. The instrument acquisition system provides a complete block of 4096 samples at once. Then, the application of different processing lengths over the complete acquired raw data has some effects on the processing due to the initialization and ending of the algorithm (see Section 4.3.2). As the size of the processed data approaches the acquired data and the measurement observation time increases, the error also increases. The maximum error ranges from 4.56% when processing 512 samples to 24.29 % when processing 2048 or 4096 samples.

#### 5.2.3. Frequency Evolution

At this stage, a possible application for our extraction algorithm is to visualize the evolution of the found frequency tones. In this sense, Figure 12 shows the temporal series of the frequency representation of the incoming signal for periods of 20.48 s (256 samples) along the complete block of acquired raw signal of 327.68 s (4096 samples).

In this frequency representation time series, variation in the measured fundamental sinusoidal tones is evident. The red curve on the right side in Figure 12 depicts the frequency decomposition of the acquired data at the beginning of the 5.46 min of measurements. This red curve presents a clear peak only and then appears to plateau without any dominant tone. However, in the adjacent black curve, it two clear peaks appear and the plateau disappears. The time distance between both curves is 20.5 s.

This capture shows the representative behavior of an ocean deep-water current. Every 20.5 s we observe how a fundamental tone exists that increases when other frequency tones are absent. Otherwise, the existence of other tones reduces the fundamental one. This observation confirms that extending the acquisition period introduces the overlapping of non-concurrent sinusoidal tones in the frequency representation, as was highlighted in previous paragraphs.

## 6. Conclusions

In this paper, we present a novel algorithm to extract the fundamental frequencies of oceanic currents using the integer arithmetic unit in a tilt-based current meter. To achieve this, a mean average filtering technique was used to determine the peaks, valleys, and zero-crossing points of the acquired raw data. The proposed algorithm exploits the selection of the filter coefficients to promote the use of additions and shifting integer mathematical operations only. In the same manner, the proposed algorithm is also focused on minimizing memory requirements.

The algorithm was evaluated for shallow- and deep-waters currents using experimental data. The conducted shallow-water experiments determined that the algorithm can correctly extract the sinusoidal tones. In this sense, the zero-crossing value error is negligible (under 0.01%). The error is below 7% in the worst case, and it is mainly produced by the sampling system. On the other hand, the algorithm was tested with real raw data from several deep-water current meters deployed at an offshore aquaculture infrastructure located in Gran Canaria (Canary Islands, Spain). In this scenario, the random nature of the traveling wave produces multiple dominant frequencies over time. The length of the processed raw data becomes the key parameter to correctly extract the fundamental frequencies of the ocean current. The maximum error obtained was below 4% when the length of the raw data was less than or equal to 512 samples. In these cases, the average error was below 1%. 

## Figures and Tables

**Figure 1 sensors-23-06549-f001:**
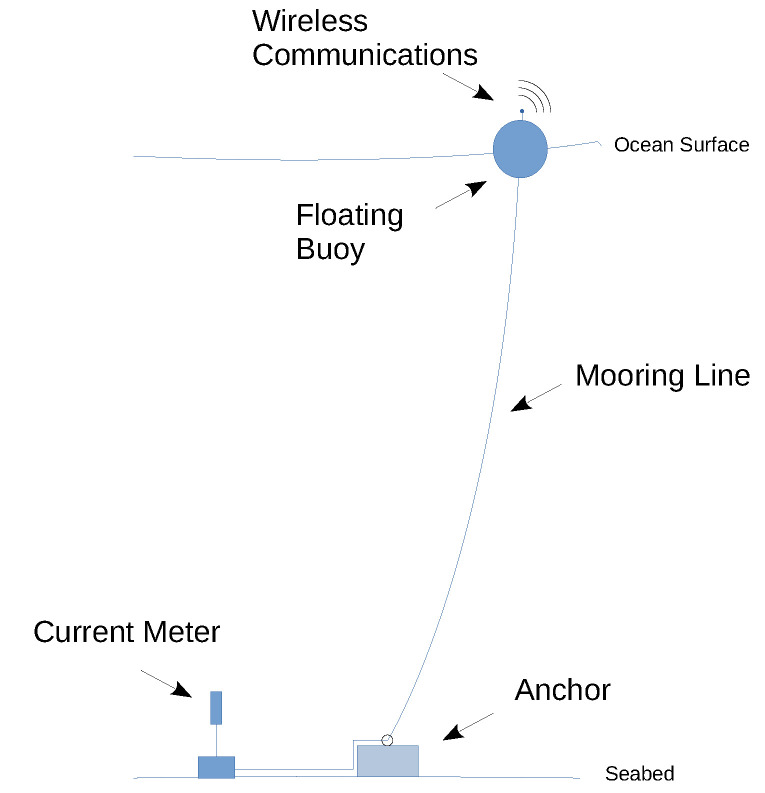
Typical ocean current meter sensor deployment infrastructure.

**Figure 2 sensors-23-06549-f002:**
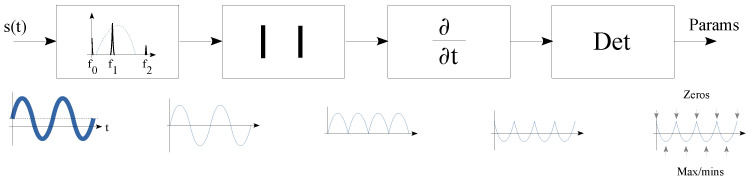
Proposed extraction algorithm block diagram.

**Figure 3 sensors-23-06549-f003:**
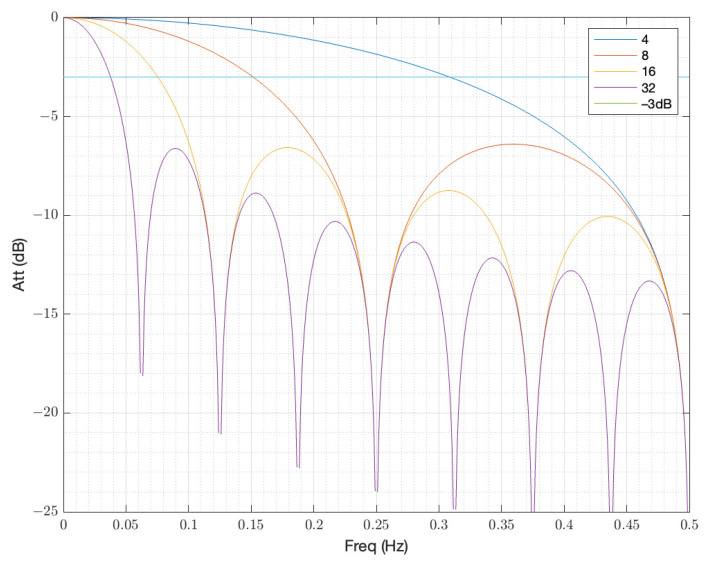
FIR filter implemented with a coefficient power of 2.

**Figure 4 sensors-23-06549-f004:**
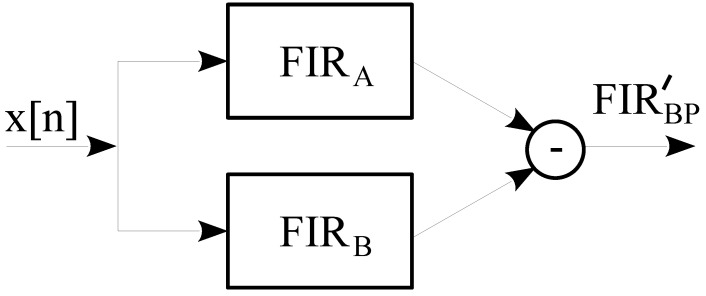
Band-pass filter implemented using two FIR filters with a coefficient power of 2.

**Figure 5 sensors-23-06549-f005:**
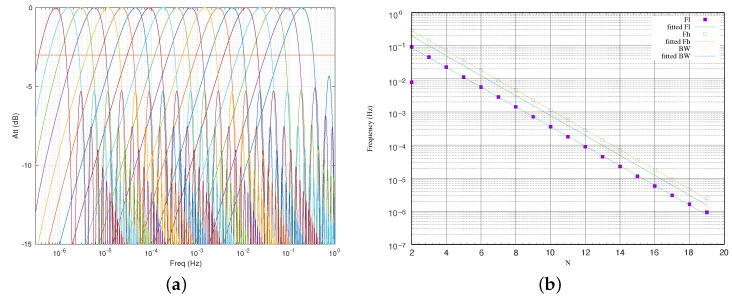
Band-pass filters implemented using two FIR filters with a coefficient power of 2: (**a**) Attenuation versus frequency; (**b**) cutoff-frequencies versus *N*.

**Figure 6 sensors-23-06549-f006:**
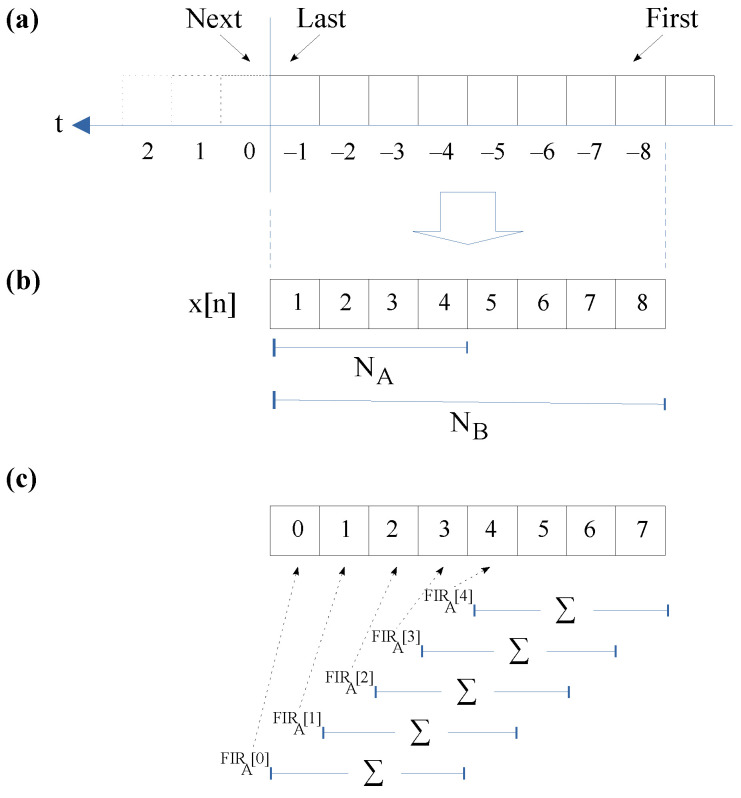
Data acquisition and FIR filter memory organization for *A* = 2 and *B* = 3: (**a**) Incoming raw data; (**b**) formulation arrangement; (**c**) microcontroller memory and FIR${}_{A}$ filter computation.

**Figure 7 sensors-23-06549-f007:**
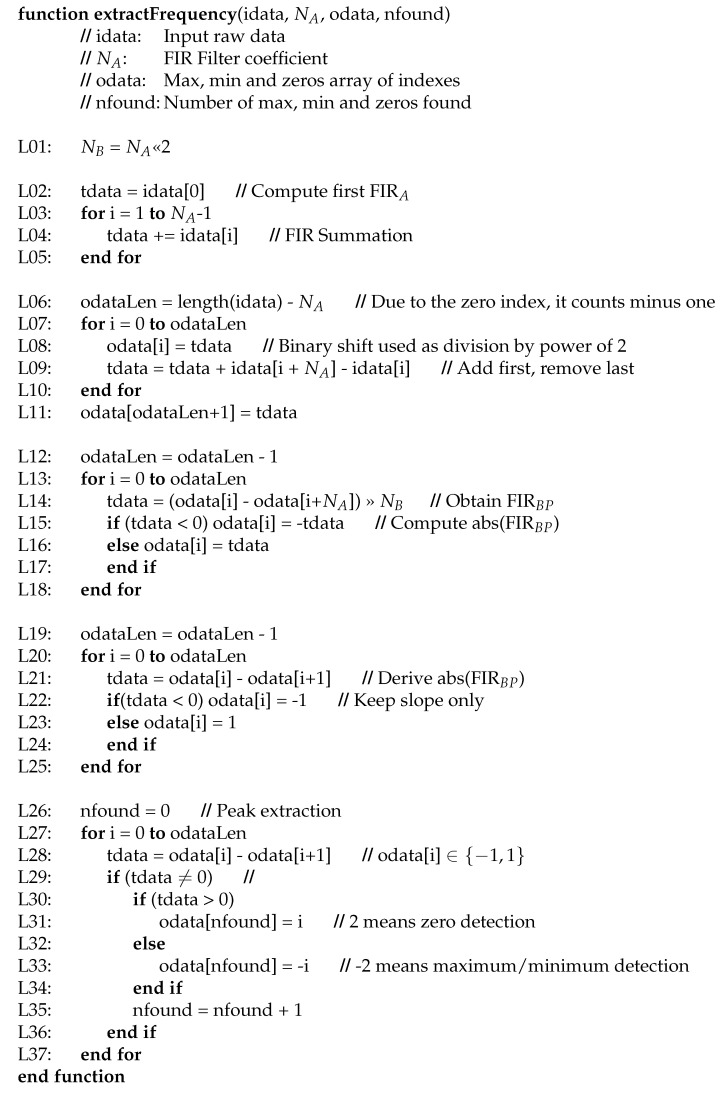
Fundamental frequency identification algorithm pseudocode implementation.

**Figure 8 sensors-23-06549-f008:**
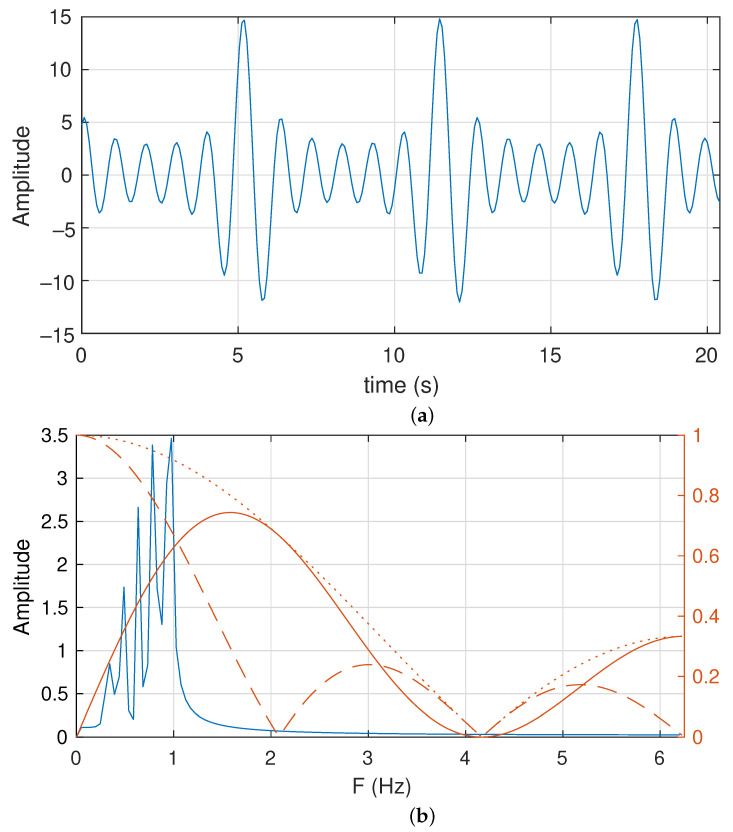
Representation of the test function F3: (**a**) time and (**b**) frequency domains.

**Figure 9 sensors-23-06549-f009:**
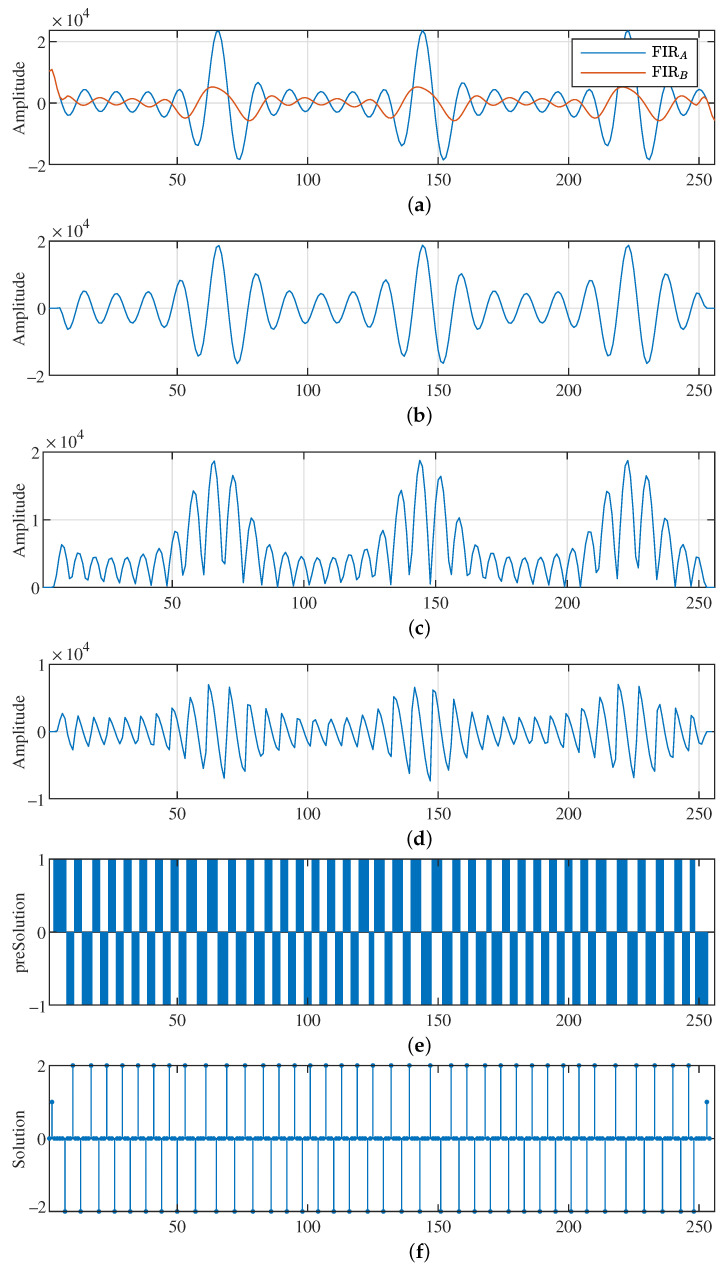
F3 signal applied to the proposed frequency extraction algorithm: (**a**) the signals after the FIR${}_{A}$ and FIR${}_{B}$ filters; (**b**) FIR${}_{B}P$ output; (**c**) phase removed; (**d**) slope tendency; (**e**) slope tendency array; (**f**) peaks, valley and zero location array. Horizontal axes are the sample index.

**Figure 10 sensors-23-06549-f010:**
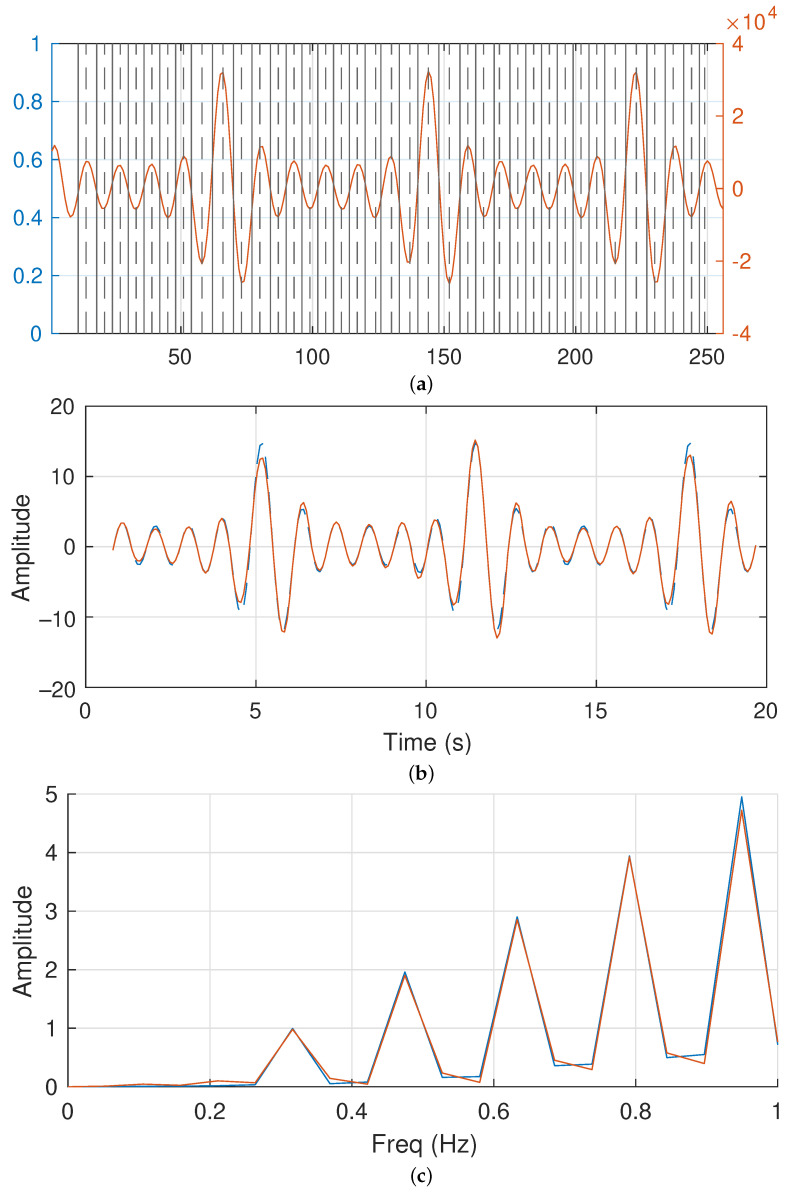
F3 signal reconstruction and comparisons: (**a**) Original sampled F3 signal and obtained peaks, valleys and zeroes; (**b**) reconstructed signal in the time domain and (**c**) frequency domain.

**Figure 11 sensors-23-06549-f011:**
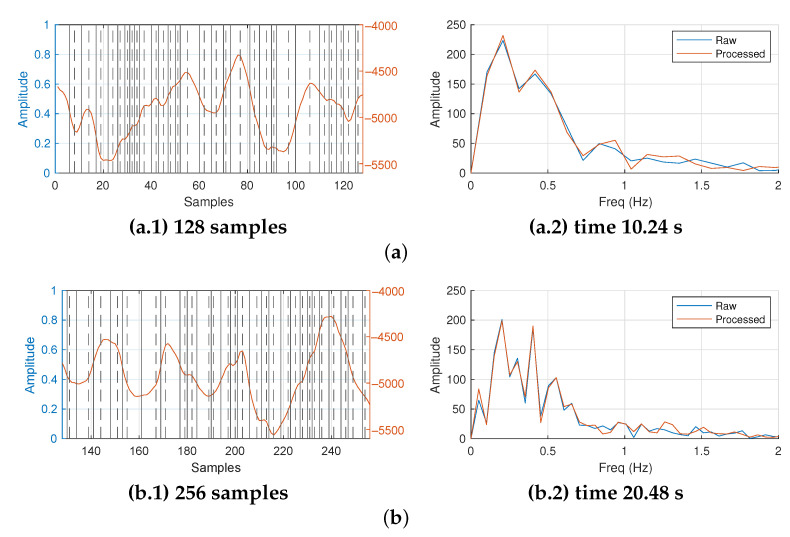
Obtained results for the real deep-water current meter measurements (raw data length): (**a**) 128, (**b**) 256, (**c**) 512, (**d**) 1024, (**e**) 2048 and (**f**) 4096 samples.

**Figure 12 sensors-23-06549-f012:**
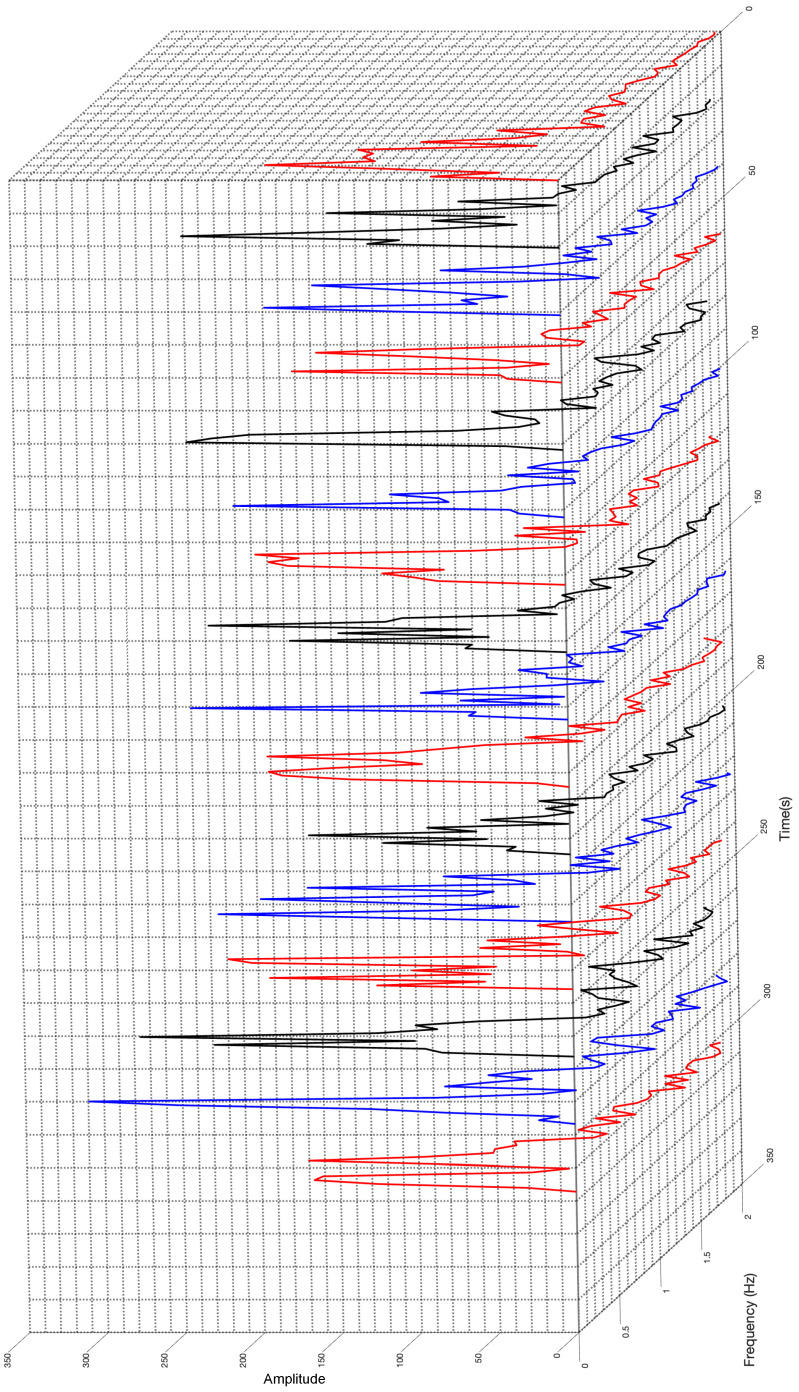
Frequency evolution along 5.46 min in intervals of 20.48 s.

**Table 1 sensors-23-06549-t001:** Published signal parameter extraction algorithms.

Reference	Year	Functions	Language	Domain	Equipment
[8]	2023	FFT, IFFT, sin, sqrt	Matlab	C	Desktop PC
[9]	2022	FFT	Matlab	C	Desktop PC
[10]	2004	EPPL, sin, integrator	Matlab	C	Desktop PC/*DSP
[11]	2012	FIR, sin, sqrt	-	R	Desktop PC
[13]	2023	Kal, sqrt, covariance	Matlab	R	Desktop PC
[15]	2017	Heuristic	C	R	Desktop PC
[16]	2020	Heuristic	-	R	Desktop PC

FFT: fast Fourier transform; IFFT: Inverse FFT; *DSP: TMS320C6711 (FPU); sin: trigonometric function; sqrt: square root function; Kal: Kalman filter; Heuristic: genetic algorithm; EPLL: enhanced phase-locked loop.

**Table 2 sensors-23-06549-t002:** Ocean current meter comparison between the research literature and commercial approaches.

Reference	Method	Location	Year	Cost	CPU	Data
[21]	Doppler	Seabed	2023	$10–20 k	NA	RAW + PC
[22]	Doppler	Seabed	2023	$10–20 k	NA	RAW + PC
[23]	Doppler	Seabed	2023	$8–15 k	NA	RAW + PC
[24]	Doppler	Seabed	2022	$50 ${}^{*}$1	Arduino	RAW + PC
[25]	Tilt	Buoy	2022	$2 k	Arduino	RAW + PC
[26]	Tilt	Seabed	2020	NA	Logger	RAW + PC
[27]	Tilt	Buoy	2014	$100	Logger	RAW + PC
[28]	Tilt	Seabed	2015	$1.1–1.5 k	NA	RAW + PC
[29]	Tilt	Mooring	2018	$50	Arduino	RAW + PC
[7]	Tilt	Mooring	2022	$50	MKL17Z256	RAW + PC

*1: Case printed in 3D PLA, without depth tests. NA: Not available. Logger: Desktop computer.

**Table 3 sensors-23-06549-t003:** Filter bank for $N_{B}$ = $N_{A}$ + 1 with a sampling frequency of 1 Hz.

$N_{A}$	$F_{l}$ (Hz)	$F_{h}$ (Hz)
2	6.13925 × $10^{-2}$	3.35325 × $10^{-1}$
3	3.03691 × $10^{-2}$	1.66480 × $10^{-1}$
4	1.51446 × $10^{-2}$	8.30954 × $10^{-2}$
5	7.56748 × $10^{-3}$	4.15297 × $10^{-2}$
6	3.78324 × $10^{-3}$	2.07627 × $10^{-2}$
7	1.89166 × $10^{-3}$	1.03812 × $10^{-2}$
8	9.45940 × $10^{-4}$	5.19069 × $10^{-3}$
9	4.73088 × $10^{-4}$	2.59546 × $10^{-3}$
10	2.36663 × $10^{-4}$	1.29784 × $10^{-3}$
11	1.18450 × $10^{-4}$	6.49043 × $10^{-4}$
12	5.93446 × $10^{-5}$	3.24641 × $10^{-4}$
13	2.97915 × $10^{-5}$	1.62439 × $10^{-4}$
14	1.50149 × $10^{-5}$	8.13390 × $10^{-5}$
15	7.62670 × $10^{-6}$	4.07887 × $10^{-5}$
16	3.93267 × $10^{-6}$	2.05143 × $10^{-5}$
17	2.08611 × $10^{-6}$	1.03765 × $10^{-6}$
18	1.16280 × $10^{-6}$	5.30776 × $10^{-6}$
19	7.01058 × $10^{-7}$	2.77330 × $10^{-6}$

**Table 4 sensors-23-06549-t004:** Results obtained for deep-water current measurements for different lengths of processed data.

Size	Max *	Avg *	Std *	Zeros	Peaks and Valleys
128	3.95	0.75	0.93	28	29
256	4.00	0.81	0.95	55	56
512	4.56	0.73	0.94	112	113
1024	5.94	0.93	1.14	230	231
2048	24.29	1.7	2.91	471	471
4096	24.29	2.02	3.12	918	919

* All values are in percentage (%).

## Data Availability

Not applicable.

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
