# Peer review of "Integer Arithmetic Algorithm for Fundamental Frequency Identification of Oceanic Currents"

_sensors, 2023, doi:10.3390/s23146549_

Round 1

Reviewer 1 Report

find the attached file. 

Author Response

Dear reviewer, thank you very much for your comments.

Your feedback help us improve our paper. Please see the attached PDF file with our responses to your comments.

Regads

Reviewer 2 Report

1. Line 103-111, mentioned here are the disadvantages of some of the algorithms to extract frequency parameters (floating point calculation cannot be applied to underwater sensors), but there are no citation here. Is this the first time to extract parameters underwater (no relevant literature introduction)? In addition, is the floating-point calculation impossible for the calculation unit of the underwater sensor?

2. The second part should focus on introducing the sampled data set.

3. Some paragraphs of the article do not add periods, such as line 111, line 608, line 625.

4. Units are not marked in some charts in the figure. What is the unit of Amplitude in Figure 9? (Other figures are similar)

5. Unlabeled information appears in some panels in the figures, as shown on the right side of Figure 9(b) and Figure 10 (a)t?(Similar to other pictures)

6. What are the variable units of Max, Avg, Std in Table 2?

7. In section 7.3 (Figure 13), it is difficult to follow what the author wants to express. It is better to convert the graph to 2D. To analyze the difference between time domain and frequency, the wavelet analysis could be added.

no other comments.

Author Response

Dear reviewer, thank you very much for your comments.

Your feedback help us improve our paper. Please see the attached PDF file with our responses to your comments.

Regards

Reviewer 3 Report

This research presents an algorithm to extract the fundamental frequencies of oceanic tidal  currents. As a tidal scientist for nearly ten years, I feel confused about this research. As far as I know, tidal frequencies are known and fixed in the global ocean, namely, there is no need to extract or determine them. Ocean currents include tidal currents and non-tidal currents. The fundamental frequencies of non-tidal currents may be indefinite and can be determined used the method proposed by this research. Tidal currents are composed of tidal constituents with fixed periods such as semi-diurnal tides (M2 with a period 12.42 hours, S2 with a period of 12 hours) and diurnal tides (S1 with a period of 24 periods). Thus, I think the authors need to revise their title, abstract and main text to clarify these questions and avoid confusions.

Other concerns:

1、The abstract is not well-written because the authors do not clearly explain their motivation of this research. Whats the problem of previous studies?

2、Line6, change is focused on to focuses on

3、Line22, change play to plays

4、The introduction is lengthy and needs to be simplified. The introductions on previous studies need to be refined.

5、Numerous short paragraphs can be merged.

English needs to be improved.

Author Response

(The authors gave the same response as above.)

Round 2

Reviewer 2 Report

I am satisfied with these replies.

Reviewer 3 Report

This paper can be accepted.